# A Generic Neural Network Approach to Infer Segmenting Classifiers for Disease-Associated Regions in Medical Images

David Schuhmacher[1,2], Klaus Gerwert[1,2], and Axel Mosig[1,2]

[1] Center for Protein Diagnostics, Ruhr-Universität Bochum, 44801 Bochum, Germany
{david.schuhmacher,gerwert,axel.mosig}@bph.rub.de
[2] Chair of Biophysics, Department of Biology and Biotechnology, Ruhr-Universität Bochum, 44801 Bochum, Germany

**Abstract.** In many settings in digital pathology or radiology, it is of predominant importance to train classifiers that can segment disease-associated regions in medical images. While numerous deep learning approaches, most notably *U-Nets*, exist to learn segmentations, these approaches typically require reference segmentations as training data. As a consequence, obtaining pixel level annotations of histopathological samples has become a major bottleneck to establish segmentation learning approaches. Our contribution introduces a neural network approach to avoid the annotation bottleneck in the first place: our approach requires two-class labels such as *cancer* vs. *healthy* at the sample level only. Using these sample-labels, a meta-network is trained that *infers* a segmenting neural network which will segment the disease-associated region (e.g. *tumor*) that is present in the *cancer* samples, but not in the *healthy* samples. This process results in a network, e.g. a *U-Net*, that can segment tumor regions in arbitrary further samples of the same type.
We establish and validate our approach in the context of digital label-free pathology, where hyperspectral infrared microscopy is used to segment and characterize the disease status of histopathological samples. Trained on a data set comprising infrared microscopic images of 100 tissue microarray spots labelled as either *cancerous* or *cancer-free*, the approach yields a *U-Net* that reliably identifies tumor regions or the absence of tumor in an independent test set involving 40 samples.
While our present work is focused on training a *U-Net* for infrared microscopic images, the approach is generic in the sense that it can be adapted to other image modalities and essentially arbitrary segmenting network topologies.

**Keywords:** Image Segmentation, Deep Learning, Segmentation-free training

## 1 Introduction

The annotation of disease-related regions is a major bottleneck for adapting machine learning in biomedical imaging. Annotations are crucially important to

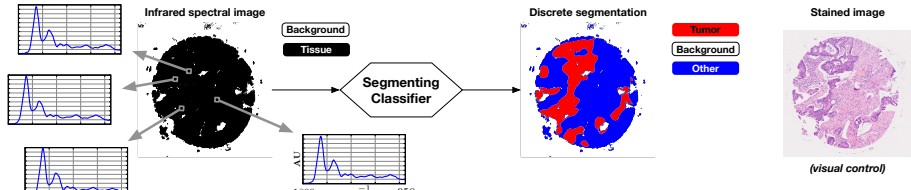

**Fig. 1.** *Principle of label-free digital pathology.* The infrared microscopic image is acquired from an unstained sample. Although the differences between different pixel spectra are barely visible by eye, it is well-established that they are highly representative for and can thus differentiate different tissue components or disease status [1, 5, 7, 11]. The spectral image can thus be used to segment disease-associated regions as illustrated, or resolve even further details of the tissue architecture [1, 11]. Our contribution deals with obtaining classifiers that can identify one disease-associated component such as tumor. In the context of label-free digital pathology, hematoxylin and eosin (H&E) stained images *(right)* are displayed mainly for visual reference and control.

train segmenting classifiers, whose relevance for medical imaging has been well realized since the introduction of *U-Nets* [17] and *SegNets* [3], not least due to the visual interpretability of segmentations by humans. However, training such segmenting classifiers requires large amounts of precise reference segmentations. Correspondingly, enormouse efforts have been made to obtain reference annotations, making annotation a labor-intensive and costly bottleneck.

As a recent and prominent example, the *Gland Segmentation in Colon Histology Images* challenge [19] provided a significant number of high-quality reference segmentations performed by a pathologist as ground truth and training data. More recently, the authors of [9] and [2] introduced crowd-based bioimage annotation systems, while the authors in [1] use an image annotation tool to obtain ground truth annotations for different components of lung tissue in 388 sample spots.

Overcoming the annotation bottleneck is particularly challenging in the context of infrared microscopic images. Infrared microscopy provides hyperspectral images of tissue samples at high spatial resolution, making it an ideal tool for resolving the tissue structure of histopathological samples and characterizing their disease status (Fig. 1). Infrared microscopy requires no staining prior to spectral image acquisition, and has been applied succesfully as a label-free digital pathology approach in increasingly large clinical studies which tackled diverse pathology related questions such as identifying tumors [13], grading colon carcinoma [11, 12] or identifying subtypes of lung carcinoma [5, 7, 1].

Our contribution aims to overcome the annotation bottleneck through an approach which trains an arbitrary segmenting neural network using annotations at the sample level only. More specifically, we assume that each sample is annotated as either a disease class such as *cancer* or a healthy class such as *cancer free*. Furthermore, we assume that each disease image contains a disease-specific region that is not contained in the healthy control samples – for example, *cancer*

samples contain a certain fraction of *tumor*, while *cancer free* spots do not contain tumor. The goal of our newly proposed *CompSegNet* approach is to train a segmenting classifier that will segment this disease-specific region.

In medical image analysis, a major advantage of segmenting networks over merely classifying networks such as plain convolutional neural networks (CNNs) is that they explicitly locate disease-specific regions, so that their output is interpretable by humans, while merely classifying CNNs are black boxes which do not provide information about the grounds on which an image was assigned to a particular class [15]. As surveyed in [8], the problem of interpreting CNN models has attracted major attention, but has been adressed in a rather post-mortem manner: A CNN is first being trained, and seeking interpretable evidence is performed only after training the network. This has been investigated using a plethora of approaches. Saliency maps [18], for example, are applicable if the images to be classified can be aligned into a unique coordinate system, as applied recently for Alzheimer's diease related whole-brain PET scans [6]. In the majority of medical image analysis tasks in radiology or pathology, where images are not alignable, techniques such as *class activation maps* (CAMs) [21] can be applied, which have been utilized recently in medical image analysis [20].

Viewed from the perspective of model interpretation, our approach shifts the interpretation problem from post-mortem to the level of training: Rather than seeking interpretable location-specific evidence in a readily trained model, we hard-wire interpretability into the target function, and the process of training optimizes interpretability of the segmentations obtained from the classifier.

## 2   Infrared Imaging Data

We use an infrared image dataset of three tissue microarray (TMA) slides involving colon carcinoma related samples. The TMA slides *CO1002b*, *CO722* and *BC051111* were purchased from *US Biomax Inc., MD, USA*, each comprising 100–200 circular spots of tissue samples obtained from more than 50 different patients. Each spot has a diameter of roughly 1 mm. Infrared pixel spectra were preprocessed using resonant Mie correction following the approach from [4], yielding the spectrum in the range between 1800–948 cm$^{-1}$ represented as a 450 dimensional vector. Each spot is represented as a $256{\times}256$ spectral image. Spectra at pixel positions not covered by sample were masked as *background* following commonly established procedures [11]. Spot labels were obtained from the annotations accompanying the data sets [3], which classify each spot into either *Malignant tumor* (**T**), *Malignant tumor (stage I–III)* (**TI–III**), *Metastasis* (**M**), *Adjacent normal colon tissue* (**NAT**), or *Normal tissue* (**N**). Following the terminology of [16], we divided the data set into three parts: 100 of the spots were selected as training set, seven spots as validation set and further fourty spots as an independent test set. Training and validation data were selected from arrays *CO1002b* and *CO722*, while the independent test set was selected

---

[3] https://www.biomax.us/tissue-arrays/Colon

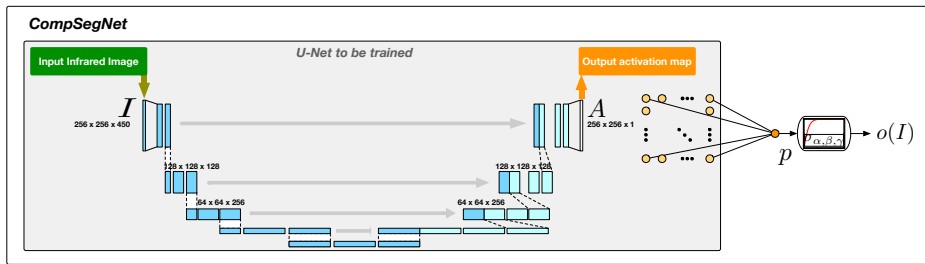

**Fig. 2.** The *CompSegNet* is a simple extension of a segmenting neural network by a pooling neuron $p$, as described in Section 3.1.

from *BC051111*. The latter array contains 180 cancer spots and 20 healthy control spots. To obtain a largest possible balanced independent test test, 20 cancer spots were chosen arbitrarily to match the 20 healthy controls, yielding our 40 spots as independent test set. In all three arrays, spots labelled as **T**, **TI–III** or **M** were used as *cancer* spots; *cancer-free* spots were selected only from those labelled as **N**.

## 3    Approach

### 3.1    Network Topology

The basic idea of our approach is to extend a segmenting neural network by accumulating the pixel activations in the output layer in a pooling neuron $p$ as displayed in Figure 2. We will refer to this extended network as a *comparative segmentation network* or *CompSegNet* for short. The specific *CompSegNet* used throughout the paper is based on the original *U-Net* topology, except the number of input channels has been extended to 450 to accomodate infrared images as input. The goal of the *CompSegNet* is to train the underlying *U-Net* in a weakly supervised manner. To this end, we assume that each pixel of the activation map is bounded within the interval $[0, 1]$, e.g. by a sigmoid transfer function, so that the pooling neuron $p$ will "count" the number of active pixels. With $p$ counting active pixels, we can specify the target function: If an input image $I^-$ is labeled as *cancer-free*, there should be zero pixels recognized as tumor, in other words, the output $o(I^-)$ of the *CompSegNet* should be zero. Conversely, if an input image $I^+$ is labeled as *cancer*, there should be a significantly large number of pixels in the activation map whose activation is close to one. In other words, $o(I^+)$ should be as large as possible.

### 3.2    Loss Function

To achieve the desired effect of training the *CompSegNet*, we design a loss function for the pooling neuron $p$ that *maximizes* the pooled activation from the

*cancer* images while *minimizing* the pooled activation from the *cancer-free* images. We assume our training data consist of spectral images $I_1, \ldots, I_M$ along with labels $\ell(1), \ldots, \ell(M)$, where $\ell(j) = 1$ if $I_j$ is a *cancer* image and $\ell(j) = 0$ if $I_j$ is a *cancer-free* image. Furthermore, we assume the transfer function $\sigma_p$ of the pooling neuron $p$ is a sigmoid-like function with an upper bound of $\alpha$, i.e., $\sigma_p(z) \leq \alpha$ for all $z$, as well as $\sigma_p(0) = 0$. After applying this transfer function, the output $o(I_j)$ of applying the *CompSegNet* output to image $j$ should satisfy the following properties:

(L1) If $\ell(j) = 1$, then $o(I_j) \to \alpha$, i.e., in a *cancer* image, a significantly large number of tumor pixels should be identified.

(L2) If $\ell(j) = 0$, then $o(I_j) \to 0$, i.e., in a *cancer-free* image, no tumor pixels should be identified.

Thus, defining the loss function $g$ as the root-mean-squared error

$$g(I, \ell) = \sum_j |o(I_j) - \alpha \ell(j)|,$$

will optimize towards properties (L1) and (L2) being satisfied.

The transfer function $\sigma_p$ for the pooling neuron $p$ needs some attention. In terms of the tumor regions to be identified, a plain sigmoid function translates to the unrealistic assumption that an "ideal" tumor sample consists of 100% tumor. In reality, it may be more realistic to assume that cancerous samples contain more than 10% tumor tissue – a value at which the sigmoid function has a steep gradient, so that a plain sigmoid target function may cause poor convergence when training the *CompSegNet*. We thus introduce a modified transfer function

$$\sigma_{\alpha, \beta, \gamma}(x) = \alpha \cdot \left(2 \cdot (1 + \exp(-\beta x / \alpha))^{-1} - 1\right) + \gamma x$$

involving three parameters $\alpha$, $\beta$ and $\gamma$. Parameter $\alpha$ specifies the minimum percentage of tumor in the cancer samples, $\beta$ is a scaling paramter, and $\gamma$ specifies a typically small asymptotic linear gain whose purpose is to avoid the vanishing gradient problem and introduce an incentive to detect more tumor in cancer samples where more significantly more than 10% of the tissue is tumor. All our experiments were conducted using fixed parameters $\alpha = .1$, $\beta = 5$ and $\gamma = .001$.

With the transfer and loss functions introduced above, training the *CompSegNet* will train the underlying *U-Net* to identify as much cancer as possible in the tumor-samples, while identifying as little tumor as possible in the tumor-free control samples. Put into even simpler terms, the *CompSegNet* is designed in a way such that training the *CompSegNet* forces the underlying *U-Net* to learn to identify the tumor regions. After convergence of the *CompSegNet* training on a sufficiently large number of images using backpropagation, the *CompSegNet* itself is no longer needed, and only the underlying *U-Net* is extracted and can serve as a readily trained segmenting network that can identify tumor regions.

*Implementation.* We trained the network described above using *RMSprop* with a mini-batch size of $k = 7$ and a learning rate of .001. Pooling neurons $p^+$ and $p^-$ use the modified sigmoid activation function $\sigma_{\alpha, \beta, \gamma}$; activation map neurons $a^+$ and $a^-$ use a standard sigmoid activation. All other neurons use the *SoftPlus* activation function. All neural networks were implemented in *tensorflow*.

## 4   Results

We evaluated our approach on the data set described in Section 2 using the network topology described in Section 3.1 using a mini-batch size of $k = 7$ and starting from random initialization. We ran 400 epochs on the test data set comprising 100 samples and selected the model with minimal loss on the seven samples from validation data set (epoch 385), requiring roughly three days of wall clock computation time using four *NVidia GeForce 1080Ti* graphics cards with a total of 44 GB of GPU memory. In the activation maps of the validation samples, a threshold was visually identified to seperate tumor from non-tumor. One and the same threshold ($.35 \cdot 255$) was applied to activation maps of 20 cancer and 20 cancer-free spots in the independent test set from TMA from slide *BC051111*. By varying a percentage threshold for relative tumor content to classify spots as *cancer* or *cancer-free*, we obtained a ROC curve with an AUC of 93.25%. The agreement between the activation maps and tumor is displayed in Fig. 3 and as Supplementary Material[4] in Sup. Fig. 2. The agreement is further supported by an average Dice score of $\mu = .42$ ($\sigma = .19$) between the thresholded activation maps and manual ground truth annotation by an independent expert for the 20 *cancer* spots underlying the ROC curve.

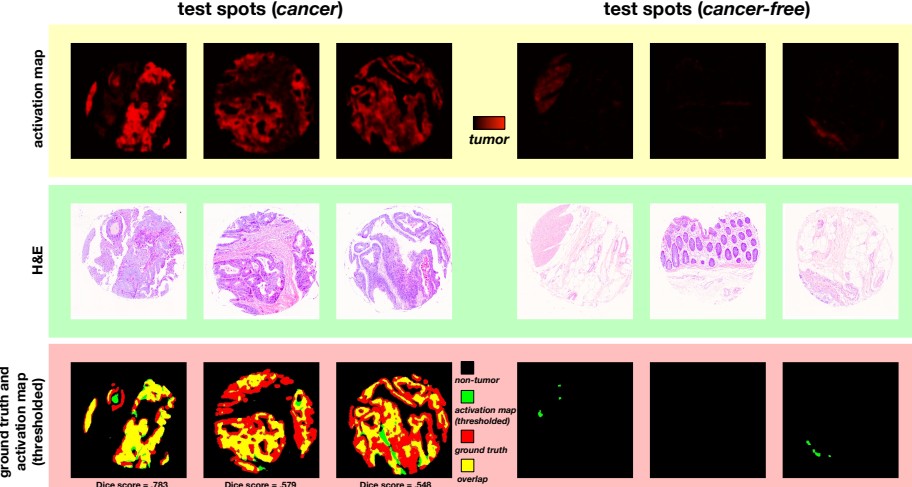

**Fig. 3.** Segmentations of six spots of the independent test data set obtained using the *CompSegNet*. Tumor regions in the three *cancer* spots agree well with the conventional staining images. Among the three *cancer-free* spots, only small amounts of false positive tumor are being detected. The complete balanced independent test set is displayed in Supplementary Figure 2.

---

[4] http://www.bioinf.rub.de/suppl/compay2019.pdf

## 5    Discussion

Our results provide proof-of-concept that segmenting classifiers can be trained on a relatively small infrared imaging data set using sample labels only, and we could demonstrate that the resulting classifier generally works on independent validation samples, despite the tissue microarray data set which certainly exhibits a higher level of heterogeneity than a typical clinical study.

Due to the generic nature of the remarkably simple *CompSegNet* approach, it suggests itself to validate the *CompSegNet* on other imaging modalities in pathology or radiology as well as on other network topologies. The approach is particularly attractive for hyperspectral modalities including variants of Raman microscocopy [14] where multidimensional imaging data inherently defy visual inspectability. For infrared microscopy driven label-free digital pathology, this immediately relieves established workflows [1, 10] from their reliance on segmentation annotations, which predominantely rely on conventional histopathological staining. It may thus infer regions that are causally linked to phenotype status that can only be determined at the whole-sample or even patient level such as *responder* vs. *non-responder*, or when contrasting differences in mutation type. It will also be relevant to investigate convergence of the *CompSegNet*, which may be improved using other target functions and other segmenting networks. Furthermore, investigating the properties of the *CompSegNet* with respect to model interpretation will be of high relevance, as well as systematically assessing how far the trained models generalize beyond the given training and validation data.

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
