# OpenReview forum: "A Generic Neural Network Approach to Infer Segmenting   Classifiers for Disease-Associated Regions in Medical Images"
_MICCAI.org/2019/Workshop/COMPAY — COMPAY 2019_

### Official Review · AnonReviewer2 · 2019-07-24
**Proof of concept study to overcome annotation bottleneck**

**Rating:** 6
**Confidence:** 4

**Review:**

This paper is a proof of concept study to overcome the segmentation annotation bottleneck. The authors use infrared microscopy data set consisting of three TMAs and propose a way to train modified segmentation networks to roughly segment cancer regions with annotations at sample level. The approach is plausible but there are caveats in its evaluation. First the authors randomly choose one of the TMAs as test set but do not specify the criteria for choosing the test set. The data set is limited but for a proof of concept study it might be good enough. The results section is considerably weak. The authors presented an approach for segmentation but are evaluating against sample level labels with an AUC value without discussing much about the ROC curve, may be because the data set is too limited. Also the dice score is very weak i.e., 0.42 which is not impressive at all. Again these results might be good enough for proof of concept study but not for a research paper and needs further refinement.

---

### Official Review · AnonReviewer1 · 2019-07-31
**weakly supervised learning with segmentation - some novelty and small scale validation (no baseline though)**

**Rating:** 7
**Confidence:** 4

**Review:**

This paper extends the weakly supervised (spot level) learning approach that is so popular at the moment to train a segmenting classifier (rather than just image level classification). This is an obvious idea, but I can't right now point to a paper that has done this in pathology and this does it in one step, so I'm going to call it novel. It turns out taking this approach is perhaps simpler than weakly supervised approaches for classification only (Fuchs et al, etc.). As the segmentation can be turned back into a classification (93% accurate) it would be interesting to see a comparison against such methods. The results for segmentation are good, but far from perfect, and there is no comparative validation against a fully supervised approach. The validation is also limited in size, and limited to one application area (IR multispectral microscopy). I generally like the use of the novel pooling layers and the idea itself, however the validation is a little weak which lets the paper down a little.

Abstract: "obtaining precise annotations" - I would disagree with the word precise as many authors have demonstrated good performance with very imprecise annotations. What you mean is you need pixel level annotations rather than spot/image level ones.

P5: "All our .. using fixed parameters" - this is again a weakness of your validation. How were the parameters determined? How sensitive are the results to changes in these parameters?

P7: "Can be trained on a relatively small data set.." You have demonstrated this for ONE EXAMPLE. This may not be the case for other examples and your claim is too strong. You should say for this type of data this is the case, but you cannot possibly say this would apply to other applications based on (relative) success on a single problem. You might want to discuss this.

---

### Official Review · AnonReviewer3 · 2019-08-22
**Review A Generic Neural Network Approach to Infer Segmenting Classifiers for Disease-Associated Regions in Medical Images**

**Rating:** 6
**Confidence:** 4

**Review:**

This paper proposes to use image-level label to infer a segmentation network for tumor delineation (pixel classification) in digital pathology.

The results are not so impressive: the Dice score is rather low and the method misses a lot of tumor regions (Fig. 3, page 6). Moreover, the dataset is rather limited. However, I recommend acceptance as a workshop paper because the idea is interesting and applied on an infrared imaging dataset.

Here are my suggestions for an extended version of this paper:
- Describe the network topology more precisely (Fig. 2 is not enough) and better explain how the spectral information is exploited.
- Provide an empirical comparison with other methods (weakly supervised and supervised pixel classifiers, as manual ground-truth is available for 20 spots) to better assess the strongness/weakness of the approach.
- Increase the size of the dataset: I think using only 20/20 spots is not well justified. I would not have reduced the independent test set from 180 cancer spots to only 20.
- Make the dataset available otherwise the study is not reproducible.
- Check that validation spots are not coming from the same patients than training set.

---

### Decision · Program_Chairs · 2019-08-20

Accept